# Extracellular Vesicle-Based Hydrogels for Wound Healing Applications

**DOI:** 10.3390/ijms24044104

**Published:** 2023-02-18

**Authors:** Andreu Miquel Amengual-Tugores, Carmen Ráez-Meseguer, Maria Antònia Forteza-Genestra, Marta Monjo, Joana M. Ramis

**Affiliations:** 1Cell Therapy and Tissue Engineering Group, Research Institute on Health Sciences (IUNICS), University of the Balearic Islands (UIB), Ctra. Valldemossa km 7.5, 07122 Palma, Spain; 2Health Research Institute of the Balearic Islands (IdISBa), 07120 Palma, Spain; 3Departament de Biologia Fonamental i Ciències de la Salut, University of the Balearic Islands (UIB), 07122 Palma, Spain

**Keywords:** extracellular vesicles, exosomes, biomaterials, hydrogels, wound healing

## Abstract

Hydrogels and extracellular vesicle-based therapies have been proposed as emerging therapeutic assets in wound closure. The combination of these elements has given good results in managing chronic and acute wounds. The intrinsic characteristics of the hydrogels in which the extracellular vesicles (EVs) are loaded allow for overcoming barriers, such as the sustained and controlled release of EVs and the maintenance of the pH for their conservation. In addition, EVs can be obtained from different sources and through several isolation methods. However, some barriers must be overcome to transfer this type of therapy to the clinic, for example, the production of hydrogels containing functional EVs and identifying long-term storage conditions for EVs. The aim of this review is to describe the reported EV-based hydrogel combinations, along with the obtained results, and analyze future perspectives.

## 1. Introduction

Torpid and slow wound healing outcomes constitute a public health problem [1] due to their high incidence, prevalence, and associated clinical assistance costs [2,3]. The effectiveness of wound healing therapies is around 50%, and all are expensive and tedious. As an alternative to standard wound dressing and conventional wound healing therapies, two main research fields are being explored to improve the risk-benefit in acute and chronic wound healing treatments [3]: extracellular vesicle (EV)-based therapies and biomaterial-based therapies [4,5,6,7].

Wound healing is a complex and multifactorial process mediated by several cellular and molecular effectors [8]. It involves the orchestration in space and time of a wide variety of cell types and molecular mechanisms (Figure 1) that takes place in four sequential and overlapping phases, including hemostasis, inflammation, proliferation, and remodeling [9]. After skin injury, vasoconstriction activates hemostasis to reduce blood flow, followed by platelet aggregation and coagulation activation to form a fibrin clot [10,11,12]. The inflammatory phase starts with the release of proinflammatory cytokines by the damaged tissues [9], promoting the migration of neutrophils, phagocytes, and macrophages to the wound [13,14,15,16]. Neutrophils are the first line barrier to avoid bacteria colonization [13,17]. Aside from combating foreign pathogens, phagocytes and macrophages also release new mediators, including vascular endothelial growth factor (VEGF) or fibroblast growth factors (FGFs), which promote angiogenesis and cell migration during the proliferation and remodeling phases [10]. Finally, the proliferation phase consists of the coordinated events of keratinocyte activation, fibroblast migration, and endothelial and macrophage cell communication that aims to orchestrate wound closure, matrix deposition, and angiogenesis, which will enhance the remodeling phase [18,19,20], where the extracellular matrix (ECM) components are reorganized to finish the wound healing process. In this last wound-healing step, the ECM spans the entire injury. Fibroblasts are the primary cell type involved in this process, promoting collagen remodeling, proteoglycans secretion, and maturation of collagen fibers [21,22,23].

Chronic wound healing is characterized by an abnormal expression of molecular and cellular effectors and by a poor immune response against pathogen colonization [10]. For instance, keratinocytes show hyperproliferation and impaired migration [18,24]. In addition, chronic wound healing is distinguished by its cellular senescence environment triggered by the pro-inflammatory environment and a high protein glycation profile in diabetic chronic wound healing [19,20,21,22,24].

Currently, the management of acute and chronic wounds consists of achieving hemostasis of the injury, preventing the colonization of microorganisms in the affected area, treating possible infections, reaching the complete closure of the wound, and, if possible, scarless wound healing. So, conventional therapies are based on applying absorbent wound dressings to remove exudate on the injury, antiseptic solutions, and oral or parenteral treatment to cure skin infections [23]. However, these therapeutic approaches have yet to fully improve the healing outcome or complete wound closure when treating ulcers with a torpid evolution. That is why it is necessary to develop new therapies for acute and chronic wounds that can improve all these aspects in managing wound healing [21]. 

During the last decade, increasing evidence suggests that cells exert their therapeutic effects in a paracrine manner, with emerging EVs as the most therapeutically potent effectors of the cellular secretome for regenerative medicine. EVs, which include exosomes and microvesicles, are spherical nanometric subcellular structures surrounded by a lipid bilayer, secreted by any cell type, and involved in cell-to-cell communication [25]. Exosomes originate from the membranous endosomal network, where intraluminal vesicles can be conducted to different destinations. Thereby, late endosomes that fuse with the plasma membrane result in the secretion of exosomes between 30–100 nm in size. In contrast, microvesicles are formed directly from the plasma membrane via budding and fission processes, giving origin to vesicles with a diameter of 50–2000 nm [26]. Their molecular cargo includes proteins, nucleic acids, and other molecules involved in cell signaling [27]. EVs function as carriers of information in the cellular environment. As dynamic systems, their content and function can be adapted according to their origin and the stimulus that promotes their release [28]. Therefore, exosomes are known to possess different molecular content and, consequently, distinctive properties depending on the cellular origin and physiological state [29,30]. EVs have shown a positive role as mediators in the wound healing process, although their molecular mechanism of action in wound healing mediation is still under investigation [5,23]. It is generally understood that EVs can interact with target cells in a variety of ways, releasing the bioactive molecules they contain and modulating different signaling pathways [31]. In this manner, EVs derived from other types of cells exhibit diverse regenerative capabilities [32]. 

On the other hand, developing novel biomaterials has been proposed as an advanced therapy in wound healing. Biomaterials constitute good candidates as substitutes for conventional therapies thanks to their intrinsic properties such as biocompatibility, improved wound healing rate, and diminution of the infection progression [7,33]. Therefore, the challenges in developing wound healing biomaterials are the maintenance of injury media, the esthetical result of the scar formation, accelerating the wound healing process, and avoiding chronic processes [7,34]. Furthermore, biomaterials must be clinically suitable, so they must be able to release active compounds over a sustained period [7,35,36]. In wound healing therapies, different biomaterials have been proposed as therapeutic options; hydrogels, electrospun nanofibers, and hybrid scaffolds stand out [35,37,38,39,40,41,42,43,44,45,46,47,48,49]. In addition, these biomaterials and EVs can activate specific signaling pathways, like collagen synthesis, cell migration, or angiogenesis [4,7]. 

In summary, in the last few years, EVs and biomaterials have been proposed as therapeutical assets in wound healing and injury repair. Both present advantages in this field and barriers that have been demonstrated to be synergically overcome in combining these two advanced therapies. This review aims to highlight the potential uses of combining EVs, which have been shown to have a relevant role in all phases of wound healing, and hydrogels, which are biocompatible, biodegradable, and allow controlled release of compounds in acute and chronic wound healing, as well as the advances that have been made in this field [50]. Therefore, in this article, we review different types of hydrogels that have been combined with EVs for treating acute and chronic wounds. We further describe the sources of EVs that are subsequently encapsulated in these hydrogels, the main results that these combinations have led to, and discuss the advantages and disadvantages of these advanced therapies.

## 2. Suitable EV-Based Hydrogels for Wound Healing Therapies

As stated above, EVs and biomaterials have achieved good results in replacing standard wound-healing therapies, which is reflected in the growing number of publications in the field. In fact, over the last decade, biomaterials combined with EVs to treat acute and chronic wound healing have been exponentially investigated (Figure 2A). 

### 2.1. Hydrogels

Hydrogels are the biomaterials preferably used to combine EVs to develop advanced therapies in wound healing. Hydrogels are three-dimensional networks of cross-linked hydrophilic macromolecules. A wide variety of polymers can be used for this purpose (Figure 2B). This heterogeneous group of hydrogel compounds includes Chitosan, Gelatin methacryloyl (GelMA), Collagen, Pluronic^®^ F-127, Hyaluronic Acid (HA), etc. [51,52,53,54,55]. Hydrogels are one of the most commonly accepted biomaterials when combined with cell secretions, as they allow the maintenance of biological activity [56]. In addition, hydrogels resemble human tissues in terms of water absorption capacity, which allows adequate release of therapeutically active particles [57].

Chitosan-based hydrogels have been the most explored in EV-based biomaterials research. Chitosan is a natural biopolymer based on amino polysaccharides, composed of an aleatory proportion of D-glucosamine and acetyl-D-glucosamine. These hydrogels have intrinsic antihemorrhagic and bioadhesive properties, among other features, such as allowing a sustained release and maintaining pH stability for EVs encapsulation [58,59,60,61,62,63,64]. In combination with EVs, chitosan-based hydrogels have achieved good results in promoting cell migration, angiogenesis, and re-epithelization [60]. A combination of the therapeutic assets has been tested in acute and chronic wound healing models, such as diabetic wound healing. For example, when pEVs are loaded in chitosan-based hydrogels with *Curcuma zedoaria* polysaccharide and silk compounds, diabetic wound healing improves the closure rate in rat models [61]. Thus, many examples have been exposed where loading hydrogels with EVs from different sources has achieved good results in wound healing research, such as EVs from bone marrow or mesenchymal stem cells (MSC) encapsulated in chitosan with collagen combinations [63]. In chitosan research, miR-126 and miR-432 from EVs have been identified as molecular effectors that achieved skin and corneal wound healing, respectively [65]. Chitosan is an example of biomaterial combined with EVs using the electrospun technique in the wound healing treatment. However, the altered membrane proteins of EVs make this biomaterial production technique difficult to reproduce in clinics [40].

Other hydrogel compositions, such as HA, are often combined with hydrogel compounds in wound healing treatment research. A natural, high-viscosity mucopolysaccharide with alternating β-(1,3)-glucuronide and β-(1-4)-glucosamine bonds that, in combination with Pluronic F127 biopolymer and with oxidative compounds such as MnO_2_, improved the glycosylated protein environment in diabetic wound healing [51]. Moreover, collagen and HA have been confirmed as EV-loaded hydrogels with ECM remodeling and restructuring capacity along with F127. Furthermore, other HA-based hydrogels have achieved gingival wound healing when combined with platelet-derived EVs [66].

GelMA, a semi-synthetic hydrogel consisting of derivatized gelatin with methacrylamide and methacrylate groups, is another common hydrogel that can be combined with EVs to treat wound healing. GelMA itself exhibits antibacterial and anti-inflammatory activity in wound media [52,64]. However, the wound healing rate has improved with GelMA hydrogels when combined with EVs [53,67]. A combination of GelMA and EVs is particularly interesting since EVs are also useful as drug delivery sites and wound-healing agents. For instance, GelMA was loaded with epidermal stem cell-derived EVs, which were loaded with VH298, a compound with wound-healing activity, achieving diabetic wound healing. However, the authors suggest that EVs densification as a drug delivery system makes it difficult for clinical translation [68]. GelMA is also combined with other biopolymers to reach hydrogels in the wound healing research field, for example, in combination with the biopolymer PEGDA with loaded HUVEC-derived EVs. In this case, GelMA/PEGDA hydrogel was used as a drug delivery system for tazarotene as an active compound, showing a full-thickness cutaneous wound on a diabetic mouse model [53]. 

Moreover, collagen-based hydrogels have been demonstrated to improve skin wound healing when combined with EVs. For instance, the collagen-chitosan combination positively affected a diabetic wound-healing rat model [69]. In addition, collagen in the hydrogel is responsible for the retention of loaded EVs. This retention permits a sustained release of EVs that are an optimum characteristic for a biomaterial in their use with EVs [70].

All these studies have achieved a synergic association between EVs and hydrogels in terms of their functional and wound-healing properties [70,71,72,73]. However, as mentioned before, EV conservation requires improvement before reaching the clinics. Furthermore, EVs are sensitive to freeze-thaw cycles and pH changes. These storage changes affect their colloidal properties. Therefore, storage conditions’ impact on EVs can affect their functionality and hinder their translation to clinics. For this reason, EV investigation requires working in the freshest conditions possible and avoiding freeze-thaw cycles. To improve storage conditions and the stability of EVs at room and physiological temperatures, EVs can be loaded into hydrogels that can maintain pH levels and colloidal properties [71,74,75]. Additionally, by regulating hydrogel properties such as porosity or degradation rate, it is possible to optimize the dose administered and EV release [76]. This allows for a localized and concentrated EV treatment by placing the hydrogel directly onto the skin area affected by the wound [77].

Another factor that can limit the clinical application of this advanced therapy is the retention of EVs by wound dressing. Absorption can result in a loss of effectiveness when the minimum effective dose is not achieved in the wound area [52,67]. To solve this problem, hydrogels with controlled and well-studied biodegradation can provide a sustained release of EVs, reaching an effective dose for a good wound healing rate.

Thus, hydrogels, for their characteristics, are the most suitable biomaterials for EV-based wound healing therapy. However, some authors have tried electrospun as a technique to coat the wound dressing with EVs. The electrospun technique presents a big disadvantage compared to hydrogels. The use of electrical potential results in a modification of protein and lipid structure and composition of the EVs’ membrane corona. Although there are good results related to electrospun biomaterials in wound healing, their combination with EVs adversely affects their function, generating a hostile environment for EVs compared with hydrogels [39,40,43,44,78].

Besides skin regeneration, corneal, gingival, and tendon healing have also shown good results with EV-based hydrogel treatments. In addition, due to the intrinsic complexity of the four overlapping phases of the wound healing process, some treatments based on EVs loaded in hydrogels failed in vitro and then achieved a good wound closure rate in vivo [63,79]. For this reason, in vivo models are the most relevant ones and should be explored to develop new acute and chronic in vivo wound healing models. Moreover, therapies based on EV and hydrogel combinations have not been tested in other types of chronic wound healing, for instance, in pressure ulcers (Figure 2C).

### 2.2. EVs Source

EVs have been shown to play a key role in several wound healing phases, such as cell migration, angiogenesis, ECM deposition, and reorganization [4]. However, one of the main variables affecting the wound healing outcome is the source from which EVs are isolated. In fact, diverse sources of EV-based biomaterials for wound healing advanced therapies have been explored (Figure 2D).

Human umbilical cord mesenchymal stem cells (hUCMSC)-derived EVs are the most studied. Originally, hUCMSC-based cell therapies were used for advanced wound care. However, using EVs may overcome many drawbacks of clinical cell-based therapies. For instance, EVs result in a reduced immune response and have less tumorigenic potential, thus improving treatment safety. Additionally, EVs can better reach their targets since EVs can pass through capillaries [80]. hUCMSC-derived EVs have been combined with different biomaterials for wound healing applications, such as Pluronic F127 hydrogel, chitosan, or HA-based hydrogels. As a result, good results have been obtained, and it has been proposed that wound closure is mediated by a high expression of VEGF, which promotes angiogenesis, and by a TGFbeta-reduced expression, which results in reduced inflammation [75]. Chitosan has been the most used material loaded with this kind of EVs, followed by GelMA and collagen hydrogels.

Other EV cell sources have also achieved good results in wound closure when combined with hydrogels: human umbilical vein endothelial cells (HUVEC), adipose stem cells, pluripotential-induced stem cells (iPSC), skin stem cells, or bone marrow. The types of hydrogels combined are diverse and report an increase in angiogenesis, a regulatory function in the inflammatory phase of wound healing, and modulation of the ECM turnover. However, there are some exceptions to these actions when EVs are combined with hydrogels. For example, by encapsulating adipose stem cell-derived EVs in antioxidant polyurethane (PUAO) hydrogels, skin infection outcomes were improved by promoting macrophage and neutrophil migration to the wound healing area [81]. These EVs have also been combined with alginate hydrogels [48], hyaluronic acid, and Pluronic F127 acid hydrogel [51]. Furthermore, epithelial mesenchymal stem cells were encapsulated in GelMA hydrogels and were used as a drug delivery system, as previously mentioned [68]. Similar results are achieved when other sources are used (Figure 2D).

These combinations have improved wound healing rates by promoting angiogenesis, re-epithelization, and ECM remodeling [14,53,54,63,67]. Some mechanisms of action that include several cell movements and molecular factors have been proposed for wound healing improvement when using EV-based hydrogels. The presence of tissue necrosis is a common feature in wounds, as is the consequent hypoxia at the injury area. Treatment with EV-loaded hydrogels can help wound closure mediated by hypoxia-inducible factor 1a (HIF-1a). Moreover, tumor necrosis factor (TNFα) and nitric oxide synthase (iNOS) are up-regulated, and TGFβ-1 is down-regulated when the wound area is treated in vivo with EV-loaded GelMA, promoting an anti-inflammatory effect. 

Micro-RNAs identified as EV cargoes also underlie the mechanisms of action for wound healing achievement. miR-223 from macrophage (M2) EVs have been proposed, in part, to be responsible for wound closure [82]. In addition, miR-432-5p can prevent collagen deposition in the ECM when it inhibits the translocation-associated membrane protein 2 (TRAM2) protein [65]. This inhibition results in a scarless wound-healing process. Moreover, miR-126 can stimulate the proliferation of dermal fibroblasts and human dermal microvascular endothelial cells in a dose-dependent way [64].

Thus, using EV-based hydrogels is effective as an advanced therapy for wound healing treatment. The role of EVs in cell-to-cell communication and the synergic activity of hydrogels is effective. It allows the combination of both to develop new active wound dressings as biomaterials and enables the control of pH and EV release.

The use of immune cell-derived EVs, such as macrophages and monocytes, in combination with hydrogels, has also been of interest in wound healing, aiming to treat and/or prevent the infection or inflammation of these injuries. In fact, their combination has improved the inflammatory phase of wound healing, promoting immune cells’ migration to the injury zone and avoiding bacteria proliferation and infection progression [55,83,84].

Platelets are another EV source being explored, as reviewed elsewhere [5]. For instance, chitosan/silk hydrogels loaded with platelet-derived EVs (pEVs) in combination with a homogeneous polysaccharide isolated from *Curcuma zedoaria* improved wound healing in diabetic rats, showing increased collagen formation and deposition, as well as angiogenesis [61]. Furthermore, the outcome of diabetic wound healing was improved when pEVs were encapsulated in an anticoagulant and antioxidant hydrogel [79]. Moreover, pEVs can also be used in regenerative medicine to treat gingival wound healing [66]. 

Another interesting EV source for wound healing purposes is honeybee *Apis mellifera* royal jelly [69]. This has been combined with type I collagen hydrogels showing a sustained fibroblast pro-migratory effect and increasing the contractile capacity of these cells, along with the inhibition of bacterial biofilm formation, improving the wound infection outcome [69].

## 3. Limitations and Future Perspectives of the Use of EV-Based Hydrogels

As described above, EV-based biomaterials have shown promising results for wound healing treatment. The combination of EVs with biomaterials reduces EVs clearance, retaining EVs at the treatment site and thus increasing treatment efficiency, aside from adding extra functionalities to the EVs effect. However, some challenges need to be addressed before their translation into clinics.

On the one hand, all chronic wound healing models are based on diabetic wound healing, which is an easily reproducible model of chronic wound healing, nevertheless having a glycated molecular environment. EV-based biomaterials should also be tested in animal models of another kind, such as chronic ulcers. Furthermore, the regenerative efficacy of EV-loaded hydrogels is tested in wound healing ratio achievement, although insufficient data has been obtained about its effectivity on infected wounds. This is why further information on the role of EV-loaded hydrogel therapy in facing skin wound infections should be performed. 

Additional challenges before clinical translation are stability, room temperature, and tissue temperature, which can impair hydrogel degradation characteristics and their release properties [83]. Moreover, EVs and hydrogels have different suitable temperatures, resulting in work being hindered by the combination of these two therapeutical assets [51]. Furthermore, working at room temperature with EV-based hydrogels can suppose an inactivation of the EVs [75]. In addition, the conservation of EVs is a barrier to their use in therapeutics. EVs are pH-sensitive and usually conserved at −80 °C when pH changes during freeze-thaw cycles can affect the number of particles obtained, the molecular cargo properties, and thus their molecular interaction mechanism [74,85]. For these reasons, it is necessary to find new strategies to improve storage and a suitable vehicle to administer them clinically. Moreover, EV isolation methods can affect their molecular cargo and functionality [25]. When EVs are combined with hydrogels, the common isolation method is ultracentrifugation; however, some groups have used size exclusion chromatography [5,66]. It is well-defined that ultracentrifugation can affect the purity of EVs obtention because of contamination with protein [86]. Another important barrier is setting the optimum gelation time when in situ gelation is desired [87]. In addition, some authors are concerned with corona protein degradation due to the gelation process [88,89]. Despite the promising results in the wound healing field, there is no standardization of the isolation method and procedures in EV research. This lack of standardization is the main barrier to translating EV therapies from bench to bedside. Therefore, the implantation of Good Manufacturing Practices (GMP) in EV production and scalation should be ensured to reach clinics.

Another important point of using EVs in clinics is their source, which can result in a differential wound healing outcome. Cell culture is the main reported obtention source of EVs for encapsulation in biomaterials. However, this source presents several disadvantages due to the economic cost or the regulatory barriers associated with it. To overcome these drawbacks, some authors propose using cell-free obtention of the EVs; for example, pEVs or *Apis mellifera* royal jelly EVs have been proposed.

Another point to be considered in EV-based biomaterials research is the need for a description of a clear mechanism of action, so more research is needed to describe and overcome regulatory barriers before reaching clinics. It is suggested that angiogenesis induced by EVs may be mediated through Erk and Akt signaling pathways, while reepithelization is triggered by the activation of yes-associated protein (YAP) [90]. These signaling pathways are involved in collagen synthesis and ECM remodeling. Each investigation performed with EV-based biomaterials brings to light the deep molecular characterization of the mechanisms underlying wound healing. This molecular mechanism is very complex, and the multiple cellular effects an EV-loaded hydrogel can produce are evidence of that. The EV-loaded hydrogels can generate crosstalk between many cell types as fibroblast, keratinocytes activation, and proinflammation mediators’ migration as macrophages [91]. In fact, EVs, through their heterogeneous components, can act on many pathways at the time, exerting a pleiotropic effect. Table 1 is shown a summary of how hydrogels and EVs can be combined in wound healing, the aim of the combination, and the molecular mechanism shown in the deep characterization. 

Since wound healing therapies aim to restore the continuity of damaged tissue, a combination of both EVs and hydrogels is very useful in regenerative medicine and should be considered to reach clinics [66]. Although results in angiogenesis, re-epithelization, and reorganization of ECM in EV-loaded hydrogel advanced therapies are promising, it should go a step further and achieve, as much as possible, a scarless regeneration of damaged tissue. This achievement may point to using EV-loaded hydrogel-based wound dressings for esthetical medicine applications. However, EVs must be studied at the ECM remodeling phase of the wound healing process to use them in this way.

## 4. Conclusions

In summary, several EV-based biomaterials have improved wound healing outcomes in preclinical studies. Both therapeutical assets present idoneous characteristics for this kind of therapy. A combination of these compounds has been tested in in vivo and in vitro models in acute and chronic wound healing treatment. EV encapsulation in hydrogels enables a sustained release and pH maintenance with enhanced exosome regenerative potential. Nevertheless, each EV source has its benefit-risk ratio, including cost and time loss in the process and the possibility of contamination. Therefore, more inference in EV-loaded hydrogels’ storage conditions should be performed before translation to clinics. Furthermore, other questions about EV-loaded hydrogels should be answered, including gelation time, optimum EV isolation methods, and the source of where EVs are obtained from. Further studies must be performed to solve these points using EV-loaded hydrogels in regenerative medicine.

## Figures and Tables

**Figure 1 ijms-24-04104-f001:**
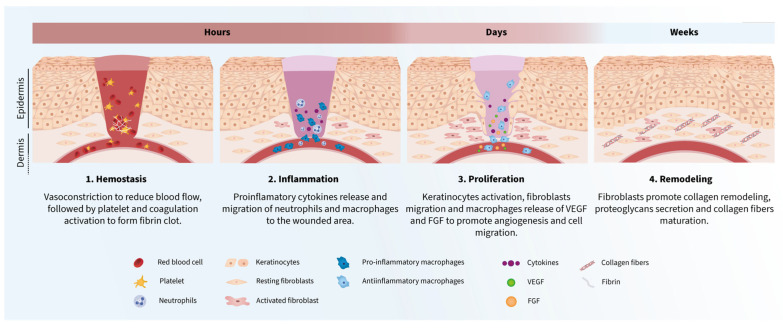
Wound healing process phases. 1. Hemostasis. Rupture of the veins adjacent to the wound area, infiltration of platelets and erythrocytes that begin the processes of platelet aggregation and coagulation that culminate in the fibrin clot formation. Tissue damage causes the release of mediators, such as proinflammatory cytokines, that will promote cell migration in later phases. 2. Inflammation. Infiltration of macrophages and neutrophils is enhanced by proinflammatory mediators to face infection and moderate inflammation. 3. Proliferation. Activation and migration of several cell types as keratinocytes and fibroblasts promoting angiogenesis and re-epithelization. 4. Remodeling. Fibroblasts orchestrate the ECM modulation for tissue regeneration.

**Figure 2 ijms-24-04104-f002:**
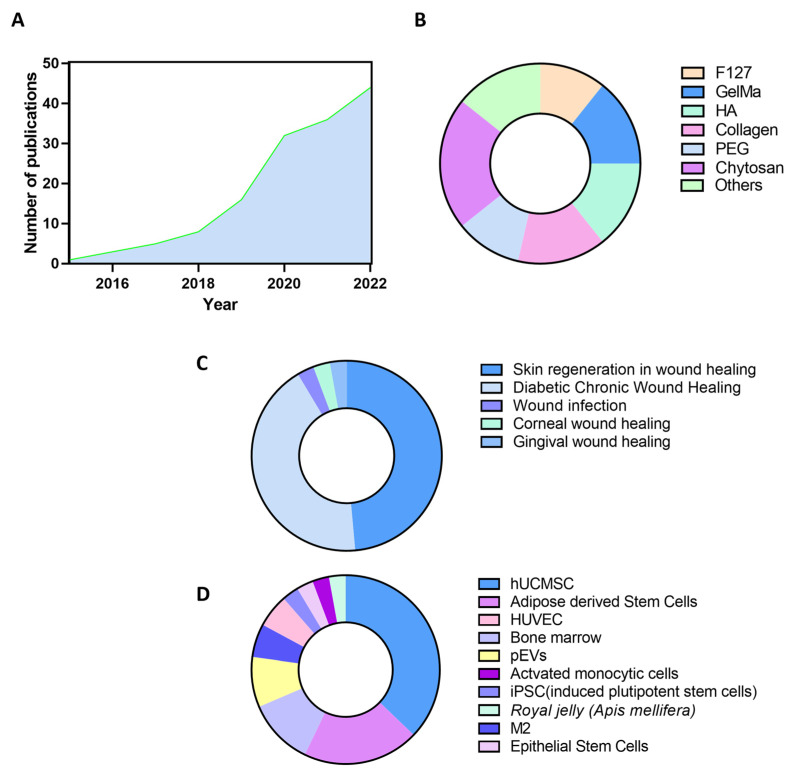
Data analysis of EV-based biomaterials research literature. (**A**) Number of publications in Pub Med with “EVs AND (biomaterials or materials) AND wound healing” per year. (**B**) Absolute frequency diagram of biomaterials and hydrogels used in EV-based biomaterials research. (**C**) Absolute frequency diagram of therapeutic use of EV-based biomaterials in acute and chronic wound healing. (**D**) Absolute frequency diagram of EV sources.

**Table 1 ijms-24-04104-t001:** Studies employing EV-based biomaterials for wound healing applications.

EVs Source		Biomaterial	Aim of Clinical Use	Cell and Molecular Mechanism of Wound Closure	Reference
**Adipose**	MSC	FHE Hydrogel	Diabetic WH	↑ Angiogenesis, collagen deposition, re-epithelization	[51]
		Methacrylate HA	WH	↑ Cell proliferation, migration, angiogenesis, and WH-related marker expression in fibroblast and endothelial cells	[92]
		PUAO and PUAO-CPO cryogel	Diabetic and infectious WH	↑ Angiogenesis, re-epithelization, fibroblasts, and keratinocytes migration	[81]
		Alginate hydrogel	WH	↑ Angiogenesis, re-epithelization, collagen deposition, and remodeling	[82]
		Pluronic F127 hydrogel	Skin WH	↑ Skin WH, re-epithelialization, expression of Ki67, α-SMA, CD31, collagen synthesis (Collagen I, Collagen III), and skin barrier proteins (KRT1, AQP3). ↓ Inflammation (IL-6, TNF-α, CD68, CD206)	[93]
		β-ChNF hydrogel	Skin WH	Metabolic, tight junction, NF-κB signaling pathways, CFD, downstream Aldolase A, and Actn2 proteins	[89]
		Polysaccharide based dressing	Diabetic WH and angiogenesis	↑ Angiogenesis, re-epithelization, collagen deposition, and remodeling	[91]
		Phospholipid-grafted PLLA electrospun micro/nanofibers immobilized	Diabetic WH	↑ Fibroblast proliferation, migration, and gene expression (Collagen I and III, TGF-β, α-SMA, HIF-1α). ↑ Keratinocyte proliferation. ↑ Expression of anti-inflammatory genes (Arg1, CD206, IL-10). ↓ Expression of pro-inflammatory genes (IL-1β, TNF-α). ↑ Cell proliferation, collagen deposition, and angiogenesis	[94]
Umbilical Cord		Recombinant Human Collagen III Protein Hydrogels	Diabetic WH	↓ Inflammatory response. ↑ Cell proliferation and angiogenesis	[71]
		Peptides based hydrogel	Diabetic WH	↑ Stimulating angiogenesis capacity	[95]
		Iron Oxide exosomes	Skin WH	↑ Endothelial cell proliferation, migration, and angiogenic tubule formation. ↓ Scar formation. ↑ CK19, PCNA, and collagen expression	[87]
		Pluronic F127 Hydrogel	Diabetic WH	VEGF and TGFβ	[75]
		Genipin crosslinked hydrogel	Skin WH	↑ Wound closure, re-epithelialization, collagen deposition, and several skin appendages	[87]
		Chitosan-SF/SA/Ag dressing	Skin WH	Antimicrobial activity. ↑ WH, retaining moisture, maintaining electrolyte balance, fibroblast proliferation, collagen deposition, angiogenesis, and nerve repair	[60]
Placental		Hyaluroran hydrogel	Scarless WH	↓ Scar tissue formation. Macrophages induction to an anti-inflammatory and anti-fibrotic (M2c) phenotype	[96]
		Chitosan—PEG Hydrogel	Skin WH	↑ Induced proliferation and vascular formation	[73]
		Collagen biomaterial	Skin WH	↓ Inflammatory responses. ↑ Muscle regeneration and vascularization	[70]
Synovium		Chitosan wound dressing	Diabetic WH	↑ Collagen deposition, angiogenesis, and re-epithelization	[64]
iPSC		Chitosan-based hydrogels	Corneal epithelium regeneration	(miR-432-5p)-mediated action	[65]
Bone marrow		CEC-DCMC hydrogel	Diabetic WH	↑ Angiogenesis, WH, and M1-type to M2-type transition of macrophages. ↓ Inflammatory effects	[54]
		Bilayered Thiolated Alginate/PEG Diacrylate Hydrogels	Scarless WH	(miR-29-b-3p)-mediated action. ↑ Angiogenesis and re-epithelization	[52]
		Collagen chitosan scaffold	WH	↑ Macrophages count, collagen deposition, and alignment	[63]
TNF-α and hypoxia treated		COF Integrated Nanoagent	Diabetic WH	↑ Anti-inflammatory M2 macrophage polarization, stabilization of HIF-1α, and angiogenesis. ↓ Oxidative injury, tissue inflammation, and bacterial infection	[97]
hEnSC		Chitosan-glycerol hydrogel	Skin WH	↑ Wound closure ability and re-epithelialization	[62]
ESC		GelMA hydrogel	Diabetic WH	↑ Angiogenesis by stabilizing HIF1α	[68]
HUVEC		GelMA/PEGDA	Diabetic WH	↑ Cell migration, angiogenesis, and exosomes/tazarotene release in the deep skin layer	[53]
		GelMA hydrogel	Skin WH	↑ Re-epithelialization, collagen maturity, and angiogenesis	[67]
M2-Macrophages		HA@MnO2 /FGF-2 hydrogel	Diabetic WH	↑ Angiogenesis, ROS depletion, collagen deposition, and remodelation	[83]
		Hydrolytically degradable PEG hydrogels	Cutaneous WH	The regulated local polarization state of Mφs and local transition from M1-Mφs to M2-Mφs within the lesion. ↑ Wound closure and increased healing quality	[84]
Monocytic cells		Electrospun nanofiber matrices	Wound infection	Bactericidal effect. ↑ HUVEC tube formation, skin cell proliferation, and migration	[55]
Platelets/PRP		HA-based hydrogels	Gingival WH	Preserved activity and functionality of platelet-derived EVs	[66]
		GelMA/SFMA/MSN-RES hidrogel	Diabetic WH	↓ Macrophage iNOS expression. ↓ Expression of TNF-α. ↑ Tube formation by hUVEC in vitro. ↑ Angiogenesis, expression of TGF-β1, Arg-1, extracellular purinergic signaling pathway-related CD73, and A2A-R	[79]
		Curcuma polysaccharide-based chitosan/silk hydrogel sponge	Diabetic WH	↑ Collagen deposition and angiogenesis	[61]
*Apis mellifera* royal jelly		Collagen Type I Hydrogel	Skin WH	↑ Fibroblast contractile capacity and migration. ↓ *Staphylococcus aureus* ATCC 29213 biofilm formation	[69]

iPSC (induced Pluripotent Stem Cells); TNF (Tumor Necrosis Factor); hEnSC (human Endometrial Stem Cells); HUVEC (human Umbilical Vein Endothelial Cells); PRP (Platelet-rich-plasma); MSC (Mesenchymal Stem Cells); ESC (Epithelial Stem Cells); WH (wound healing); PUAO (antioxidant polyurethane); CPO (calcium peroxide); β-ChNF (β-chitin nanofiber); PLLA (poly-l-lactic acid); PEG (Polyethylene glycol); COF (covalent organic framework); GelMa (Gelatin Methacryloyl): PEGDA (Polyethylene glycol diacrylate); HA (hyaluronic acid); FGF (fibroblast growth factor); SF (silk fibroin); SA (stearic acid); IL (interleukin), α-SMA (Alpha Smooth Muscle Actin); KRT (keratin); AQP (aquaporin); NF-κB (nuclear factor kappa B); CFD (Complement Factor D); Aldoa (Aldolase A); Actn2 (Alpha-actinin-2); TGF (Transforming growth factor); HIF (Hypoxia Inducible Factor); CK (Cytokeratin); PCNA (proliferating cell nuclear antigen); VEGF (vascular endothelial growth factor); iNOS (nitric oxide synthase); Arg (arginase); A2A-R (adenosine A2A receptor); EV (extracellular vesicles); ROS (reactive oxygen species); CEC (carboxyethyl chitosan); DCMC (dialdehyde carboxymethyl cellulose); SFMA (silk fibroin glycidyl methacrylate); RES (resveratrol); MSN (mesoporous silica nanoparticles). ↑: increase. ↓: decrease.

## Data Availability

Data sharing not applicable.

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
