# Peer review of "Extracellular Vesicle-Based Hydrogels for Wound Healing Applications"

_ijms, 2023, doi:10.3390/ijms24044104_

Round 1

Reviewer 1 Report

In this article, entitled “Extracellular vesicles-based biomaterials for wound healing applications”, the authors provide comprehensive reviews of current therapies with biomaterials and extracellular vesicles (EVs) for wound repair. There are some minor comments.

1. Overall, it is unclear how EVs affect the healing process. The authors show various sources of EVs (MSCVs, HUVEC, iPSC, and immune cells) and their effects (re-epithelization, angiogenesis, and anti-inflammation). However, it is still unclear what is inside the EVs specifically and how these molecules work during the healing process. The authors said there are RNAs and proteins, but it is vague. Exactly which RNAs and proteins are present in EVs? Except for the miRNAs, the mechanisms are elusive, and it looks like a "cure-all" drug.

2. It would be beneficial if there were more detailed explanations about the characteristics of EVs. The authors said EVs include exosomes and microvesicles. It would be better to illustrate more basic information about EVs, such as differences in size and how these are made in the cells.  In addition, it is necessary to explain action mechanisms: how they work in the injured tissue. For example, macrophages or other immune cells engulf EVs? EVs are ruptured in vivo and released inside molecules to the wound bed?

3. The authors mentioned "biomaterials," but the majority are hydrogels. Are there other non-hydrogel biomaterials that can be used with EVs? Otherwise, are there specific reasons why hydrogel is the main partner for EVs delivery?

4. The authors said the combination of EVs and biomaterials would be synergistic. They mentioned that hydrogels retain EVs better and maintain pHs well. Are there additional advantages in addition to the "delivery"? Is there any advantage in wound healing when EV is mixed into biomaterials rather than direct EV administration or EV expressing-cell transplants?

Author Response

  1. Overall, it is unclear how EVs affect the healing process. The authors show various sources of EVs (MSCVs, HUVEC, iPSC, and immune cells) and their effects (re-epithelization, angiogenesis, and anti-inflammation). However, it is still unclear what is inside the EVs specifically and how these molecules work during the healing process. The authors said there are RNAs and proteins, but it is vague. Exactly which RNAs and proteins are present in EVs? Except for the miRNAs, the mechanisms are elusive, and it looks like a "cure-all" drug.

In agreement with the reviewer’s comment, we have added more information about what is known on EVs content, highlighting the importance of EVs cell source for their final effect. 

  1. It would be beneficial if there were more detailed explanations about the characteristics of EVs. The authors said EVs include exosomes and microvesicles. It would be better to illustrate more basic information about EVs, such as differences in size and how these are made in the cells.  In addition, it is necessary to explain action mechanisms: how they work in the injured tissue. For example, macrophages or other immune cells engulf EVs? EVs are ruptured in vivo and released inside molecules to the wound bed?

In agreement with the reviewer’s comment, additional explanation on EVs have been added to this new version of the manuscript. 

  1. The authors mentioned "biomaterials," but the majority are hydrogels. Are there other non-hydrogel biomaterials that can be used with EVs? Otherwise, are there specific reasons why hydrogel is the main partner for EVs delivery?

We agree with the reviewer, in accordance, we have changed the term biomaterials in the title and in many other parts of the manuscript for hydrogels. Is true that other biomaterials can be combined with EVs, but so far, in the field of wound healing applications, hydrogels have been the biomaterial most widely used. A comment on the advantadges of using hydrogels has been added in this new version of the manuscript. 

  1. The authors said the combination of EVs and biomaterials would be synergistic. They mentioned that hydrogels retain EVs better and maintain pHs well. Are there additional advantages in addition to the "delivery"? Is there any advantage in wound healing when EV is mixed into biomaterials rather than direct EV administration or EV expressing-cell transplants?

In agreement with the reviewer’s comment, a comment on the advantadges of using hydrogels has been added in this new version of the manuscript. 

Reviewer 2 Report

This manuscript reports the most recent advances of extracellular vesicles-based biomaterials for wound healing applications. This work is meaningful, and the manuscript is well-organized and written. Before considering this manuscript for publication, the authors should consider the following points in any revision as follows:

1.     I suggest the authors to briefly describe the contents of the review in the last part of the Introduction.

2.     The type and information of the references should be standardized and complete. The page number were missed for lots of references.

3.     Some recent references should be cited, such as Engineering Reports 2020, 2, e1214; Advanced Functional Materials 2021, 31, 2105718; Chinese Chemical Letters 2022, 33, 5030-5034; Chinese Chemical Letters 2022, 33, 1880-1884; Materials Today Bio 2022, 16, 100429.

Author Response

1.     I suggest the authors to briefly describe the contents of the review in the last part of the Introduction. 

In agreement with the reviewer’s comment, in this new version of the manuscript we have added a brief description of the contents of the review in the last part of the introduction. 

2.     The type and information of the references should be standardized and complete. The page number were missed for lots of references. 

We have carefully checked al the information given in the references in this new version of the manuscript. We apologize for this mistake. 

3.     Some recent references should be cited, such as Engineering Reports 2020, 2, e1214; Advanced Functional Materials 2021, 31, 2105718; Chinese Chemical Letters 2022, 33, 5030-5034; Chinese Chemical Letters 2022, 33, 1880-1884; Materials Today Bio 2022, 16, 100429. 

Respectfully to the reviewer’s comment. The references that the reviewer suggest are important articles on hydrogels for wound healing applications, but in this review we have only selected those articles that combine the use of hydrogels with the use of EVs, therefore, these references have not been added to the new version of the manuscript.